# Foreign Body Ingestion in Children: Epidemiological, Clinical Features and Outcome in a Third Level Emergency Department

**DOI:** 10.3390/children8121182

**Published:** 2021-12-15

**Authors:** Antonio Gatto, Lavinia Capossela, Serena Ferretti, Michela Orlandi, Valeria Pansini, Antonietta Curatola, Antonio Chiaretti

**Affiliations:** 1Institute of Pediatrics, Fondazione Policlinico Universitario A. Gemelli IRCCS, 00168 Rome, Italy; valeria.pansini@policlinicogemelli.it; 2Institute of Pediatrics, Fondazione Policlinico Universitario A. Gemelli IRCCS—Università Cattolica Sacro Cuore, 00168 Rome, Italy; laviniacapossela@gmail.com (L.C.); serena.ferretti01@icatt.it (S.F.); michela.orlandi01@icatt.it (M.O.); c.anto91@hotmail.it (A.C.); antonio.chiaretti@unicatt.it (A.C.)

**Keywords:** foreign bodies, ingestion, endoscopy

## Abstract

Ingestion of foreign bodies is a frequent pediatric cause of access to the Emergency Department (ED). The aim of this study was to determine the epidemiological and clinical features of pediatric patients with a diagnosis of foreign body ingestion and to identify the factors associated with an urgent invasive procedure or hospitalization. This is a retrospective study conducted on a population of 286 pediatric patients (0–17 years) evaluated for foreign body ingestion at the Pediatric ED of “Fondazione Policlinico Universitario A. Gemelli, IRCSS” between October 2014 and June 2019. Data concerning age and gender, underlying diseases, type of foreign body, symptoms and signs, instrumental tests, specialist visits, treatment and outcome were analyzed. The majority of foreign bodies were coins (23%). Symptoms recurred in 50% of the foreign bodies with esophageal localization and between the 92 (32%) patients with symptoms the most common was vomiting (7%). X-rays was performed in 61% of patients. Among all patients, 253 patients (88.8%) had been discharged, 21 (7%) had been hospitalized, and four (1.4%) were sent to an outpatient facility. Besides, 17 (5.9%) patients had been transferred to the Observation Unit. Of the hospitalized patients (21 (7.3%)), clinical observation was performed for 57% and endoscopic procedure for 45%. Our data confirm that the ESPGHAN-ESGE guidelines application prevents interventions that are not necessary, avoiding diagnostic and therapeutic delays.

## 1. Introduction

Ingestion of foreign bodies is a common cause of access to the Emergency Department (ED) in pediatric patients, with up to 75% of cases occurring in children under 4 years of age. According to [1,2], 98% of ingestions are accidental. The foreign bodies ingested are mostly found in the domestic environment and their morphology and size are variable.

In 80–90% of cases, the foreign body passes without complications and is evacuated with feces within a few days, and 10–20% of cases may require endoscopic removal because the foreign body does not pass easily or because it is potentially harmful. Less than 1% may require surgery [3,4,5].

The increasing prevalence of smaller, more technologically advanced toys in the household has resulted in an increased exposure to higher voltage batteries and powerful magnets that carry a high incidence of morbidity and mortality [6].

In most cases, children are asymptomatic and come to the pediatrician’s attention because ingestion is witnessed [7,8]. If there are symptoms, they are related to the location of the foreign body in the gastrointestinal tract and its characteristics. Patients with a foreign body in the esophagus may complain of dysphagia, sialorrhea, cough, hematemesis, globus sensation, thoracic cluttered sensation or respiratory symptoms due to the compressive effect on the trachea. Patients in whom the foreign body transits into the stomach are typically asymptomatic, although large objects may cause pyloric obstruction and cause vomiting and/or refusal to feed [9]. Similarly, patients in whom the foreign body passes through the intestine are usually asymptomatic; ileocecal valve retention rarely occurs, with possible complications such as obstruction, perforation and peritonitis. A careful medical history and physical examination are essential for the diagnosis [10].

The recent guidelines of the European Society of Gastroenterology, Hepatology and Pediatric Nutrition (ESPGHAN) clearly describe the indications and timing of endoscopic surgery [11,12].

We describe a population of pediatric patients with the diagnosis of foreign body ingestion, to determine the epidemiological and clinical characteristics of these patients and to analyze the frequency of factors and circumstances associated with the necessity of an urgent invasive procedure or hospitalization.

## 2. Materials and Methods

This is a retrospective study conducted on a population of pediatric patients (0–17 years) evaluated at the Pediatric Emergency Room of the “Fondazione Policlinico Universitario Agostino Gemelli, IRCSS” between October 2014 and June 2019 with the diagnosis of foreign body ingestion.

Patients were identified from the hospital computerized clinical record (GIPSE^®^) by searching for the keywords “Foreign body ingestion”, “Foreign body”, “Foreign body obstruction” for all patients admitted to the ED. Clinical and demographic data were collected by pediatric specialists after being trained in data collection by a form developed specifically for the study. Reports of the instrumental examinations and endoscopic procedures performed were extracted by reviewing the text of the procedure. Data concerning age and gender, underlying diseases, type of foreign body, symptoms and signs, instrumental tests, specialist visits, treatment and outcome of the intervention were collected and analyzed.

The aim of the study is to determine the epidemiological and clinical characteristics of these patients and to identify the factors and circumstances associated with an urgent invasive procedure or hospitalization. The diagnostic–therapeutic management of the observed cases was also evaluated in comparison with the most recent clinical–therapeutic recommendations [12]. 

Data concerning categorical variables are expressed in numbers and percentages. Continuous variables are expressed as mean ± standard deviation.

Comparison between groups of categorical variables was performed using the Yates corrected chi-square test. A *p* value < 0.05 was required for statistical significance.

## 3. Results

During the study period in the Pediatric Emergency Room of the “Fondazione Policlinico Universitario Agostino Gemelli, IRCSS”, 286 children with a diagnosis of suspected or certain ingestion of a foreign body were recorded; 162 patients were male (56.6%) and 124 female (43.4%). Referring to the urgency of the patients’ examination at the triage, 65% (187) were admitted with a green code (uncritical patient, low priority access to care), 35% (99) with a yellow code (moderately critical patient, quick access to treatment).

In almost all cases (98.3%, 281), the ingestions occurred as accidental episodes at home. Only in the case of one female patient of 14 years (0.3%) was the ingestion voluntary. Only four (1.4%) patients were affected by neuropsychiatric conditions. Between the ages of 0 and 3 years we observed 163 patients (57%), with a high prevalence between 1 and 2 years (115 (40%)). The average age observed was 4 years, with a standard deviation of 3.3. The smallest patient was a 2 month-old infant.

The foreign bodies described in the anamnesis showed a wide variability in terms of morphology and size (Table 1).

Analyzing signs and symptoms, 194 patients (68%) were asymptomatic. Between the 92 (32%) patients with symptoms, the most common are shown in Table 1.

Symptoms recurred in 50% (4/8) of the foreign bodies with esophageal localization, in 7% (2/30) of the foreign bodies with gastric localization and in 17% (7/40) of foreign bodies located beyond the pylorus: this correlation between the symptoms and the location of the foreign body found on X-ray was statistically significant (χ^2^ = 8.58, *p* = 0.0137).

Sharp objects were associated with symptoms in 47% of cases; in particular, patients who had ingested fish bones were symptomatic in 100% of cases.

Coins and other blunt objects (buttons, marbles, beads) were associated with symptoms in 22% of cases.

The relationship between foreign body morphology, pointed or blunt, and the presence or absence of symptoms was statistically significant (χ^2^ = 7.3356, *p* = 0.00676). 

Physical examination was normal in 272 patients (95%); anomalies were found in 14 cases (5%), 7/14 (50%) had superficial lesions of the oral cavity mainly associated with the ingestion of glass fragments, 2/14 (14%) pain in the epigastric region, 1/14 (7%) slightly globous abdomen and 1/14 (7%) a small longitudinal lesion of the perianal region.

Considering instrumental tests and specialist visits, 175 (61%) patients underwent X-rays. Only one patient underwent two projections X-rays, while the others only antero-posterior projection.

In 100 patients (57%), the foreign body was visualized. In the remaining 22 cases (22%), the foreign bodies had an undefined localization. Among patients whose foreign body was radiographically viewed, 83 (83%) were asymptomatic and 19 (19%) had symptoms. In 75 patients (43%), the foreign body was not visible. A second examination was performed in nine patients (5%) undergoing X-rays. Specialist visits were requested in 33 cases (12%): pediatric surgery consultation in 16 (6%), digestive endoscopy consultation in eight (3%), Poison Control Center consultation in five (1.8%), because of the ingestion of potentially toxic materials, and Ear-Nose-Throat (ENT) consultancy in seven cases (2.4%). 

Data about treatment and outcome showed that 253 patients (88.8%) had been discharged, 21 (7%) had been hospitalized and four (1.4%) sent to an outpatient facility. 

Besides, 17 (5.9%) patients had been transferred to the Observation Unit (OU) [13]. The average time spent in OU was 12 h, 7/17 (41%) of these patients underwent endoscopic surgery for foreign body removal, 5/17 (29%) patients had been discharged, 5/17 (29%) patients in OU were hospitalized: two of them subsequently underwent endoscopic removal of the foreign body and three were clinically observed. The hospitalized patients were 21 (7.3%): 12/21 (57%) were clinically observed and 9/21 (45%) underwent endoscopic procedure.

Among hospitalized patients, only six (29%) were symptomatic, no statistically significant correlation was demonstrated between the presence of symptoms and the need for hospitalization (χ^2^ = 0.049, *p* = 0.9440). Out of all patients, endoscopic removal was performed in 17 (6%) cases. 

Among the 17 patients who underwent endoscopy, only seven (41%) presented with symptoms. No statistically significant correlation was demonstrated between the presence of symptoms and the need for endoscopic surgery (χ^2^ = 0.0848, *p* = 0.7708). 

The types of foreign body ingested were various: in 5/17 cases (29%) coins, in five (29%) sharp objects, in two (12%) button batteries, in two (12%) clothespins > 2.5 cm, in one (6%), respectively, food bolus, a plastic object of 4 cm and a metal object. In 9/17 (53%) cases the endoscopy performed identified and removed the foreign body, in 8/17 (47%) cases the endoscopies did not visualize the foreign body (Table 2).

Of the total 253 patients discharged (88.8%), nine (3.5%) were admitted to the ED a second time. Four patients (1.5%) had a second ED access 20 days later, reporting failure to evacuate and one (0.4%) 50 days later for the onset of abdominal pain. In these cases, a control X-ray was performed: only in one patient was the persistence of the foreign body documented, with progression compared to the location observed at the previous access. 

## 4. Discussion

Ingestion of foreign bodies is a common cause of evaluation in the Pediatric Emergency Department. In our series, most patients at triage were admitted with a green code (uncritical patient, low priority access to care).

Ingestions occur in almost all cases as accidental events in children without underlying pathologies [14,15]: in our series, only 1.4% of the patients had neuropsychiatric disorders.

Most patients are between 0 and 3 years old, with the highest incidence between 1 and 2 years. The incidence gradually decreases from 6 years onwards and it is rare to observe ingestion episodes between 11 and 17 years. Since ingestions usually occur accidentally in otherwise healthy children, age is the main risk factor. The most frequent foreign bodies are coins, which represent almost a quarter of total cases. Other foreign bodies are extremely variable in morphology and size: small and blunt objects such as coins, buttons, marbles, beads; vulnerable objects, pointed or large (large or long); button batteries; food boluses, fish bones and bones.

Symptoms were mainly abdominal and most frequently were vomiting or retching, dysphagia and abdominal or epigastric pain. The presence of symptoms is related to the location of the foreign body: esophageal foreign bodies are more frequently associated with symptoms [16], while patients with gastric or intestinal foreign bodies are more frequently asymptomatic.

As for the type of foreign body, there is a correlation between the ingestion of sharp objects and the presence of symptoms. Indeed, considering only patients who ingested sharp objects, the percentage of symptomatic rises from 35% to 47%. Among patients with symptoms, physical examination was negative in 95% of them. 

The imaging techniques allow the confirmation of the presence of the foreign body and the evaluation of its position, size, and shape: the guidelines recommend performing an X-ray in two projections (anteroposterior, lateral) of the chest and abdomen in all patients with suspected ingestion of a foreign body, even if asymptomatic [17]. The examination allows the visualization of most radiopaque foreign bodies, although radiolucent objects are not uncommon, so the reliability of this type of investigation is not absolute. The choice to perform exams should not be based only on the clinical presentation: in our series only 19% of the foreign bodies confirmed by the X-ray were associated with symptoms. The most frequently prescribed exams in our study were the chest and abdomen X-rays, in accordance with the guidelines. Furthermore, almost all of the radiographs in our population were obtained only in the frontal projection, while the guidelines recommend always performing the exam in two projections [18].

It is essential to establish whether the patient requires surgery or not, to minimize the risk of possible complications. In asymptomatic patients with gastric or intestinal foreign bodies and without risk elements, home observation is generally indicated: the child must maintain a regular diet and the stools must be checked for evacuation of the foreign body, which in most cases occurs in 4–6 days but can take up to 4 weeks for small blunt items. 

The presence of symptoms must be considered as an alarm signal since they can indicate complications, but at the same time their absence is not predictive of a lower severity case [19]. We observed that, in our series, the presence of symptoms is not significantly correlated with hospitalization or endoscopic removal. Thus, a decision on the intervention in an ingestion case could not be based only on the clinical condition of the patient, but it is very important to consider the location and the characteristics of the foreign body, in order to identify possible risk factors for complications [10,20,21,22].

In our series, 88.8% of patients did not require any intervention and were discharged. On the other hand, 7% of patients required hospitalization for a vulnerable foreign body in a region not evaluable with endoscopy, or a sharp radiolucent foreign body or for observation to rule out complications that required endoscopic surgery. 

According to the literature, endoscopic removal of the foreign body is necessary in a small percentage of patients [11,12,23]. In symptomatic patients with a foreign body in the esophagus, removal is indicated within 2 h of presentation. In asymptomatic patients, removal is, however, indicated if the foreign body has been stuck in the esophagus for more than 24 h (or for an unknown period). Guidelines recommend urgent endoscopic removal of blunt objects from the stomach if the patient is symptomatic or if the foreign body is greater than 2.5 cm in diameter or 6 cm in length. Some types of foreign bodies deserve special attention because they determine a higher risk of complications: if ingested, button cell batteries can cause caustic lesions, ulcerations and, if persisted for a long time, perforation. Due to the increased likelihood of sequelae, guidelines recommend removal within 2 h for multiple button cell batteries in the esophagus [24]. A coin cell battery in a symptomatic patient, regardless of location, requires emergency endoscopic removal [25,26].

Foreign bodies with sharp edges can damage the mucous membrane of the digestive tract, especially if localized in the esophagus. Many sharp objects are radiolucent; therefore, the positive predictive value of the radiographic examination in some cases is low [14,25]. Guidelines recommend the emergency endoscopic removal of all sharp objects located in the esophagus, stomach and proximal duodenum.

In our study, the indication for endoscopic removal was given in 6% of overall cases with the diagnosis of suspected esophageal foreign bodies. Among endoscopies performed, 53% successfully removed the foreign body. However, 47% of endoscopies did not visualize the foreign body: we observed that, in 63% of these patients, the endoscopy was not conclusive. In the remaining 37% of the cases in which the endoscopy did not lead to the removal of the foreign body, the objects were radiolucent and it was impossible to locate them, thus the decision for the intervention was made on the basis of the information that emerged from the anamnesis about the characteristics of the object. In addition, it is important to point out that if foreign-body impaction lasts for more than 24 h, there is a significant increase in the incidence of complications [27,28,29].

The ESPGHAN-ESGE guidelines [10] clearly indicate the management and decision timing in cases of ingestion in a child. Their application allows the avoidance of interventions that are not really necessary or the avoidance of delays. Performing endoscopy only when indicated means not subjecting the child to unnecessary physical and psychological stress, avoiding the risks associated with the intervention and saving in terms of health resources.

## 5. Conclusions

Our study confirms that the ingestion of foreign bodies is a frequent pediatric cause of evaluation in the Emergency Department. According to the literature and most recent guidelines, the evaluation and management of these cases must be quick but thoughtful, in particular with reference to the performance of invasive procedures, because of the age of patients and the related complications.

## Figures and Tables

**Table 1 children-08-01182-t001:** Foreign bodies and symptoms.

Type of Foreign Body	n	%
Food (impact)	11	3.8
Foreign bodies *	204	71.3
Sharp objects **	36	12.6
Batteries	12	4.2
Other ***	23	8.1
Symptoms	n	%
Vomit	20	7
Dysphagia/pharyngodynia	19	6.6
Abdominal/epigastric pain	11	3.8
Drooling	12	4.1
Gagging	9	3.1
Dyspnea	9	3.1
Cough	7	2.4
Other	18	6.2

* Coins, Marbles, Beads, Jewelry (earrings, rings, pendants), Hair clips, Rubber objects, Paper and cardboard. ** Fish bones, Plastic objects/fragments, Shards of glass, Metal objects/fragments, Toothpick, Aug, Screws and nails, Thumbtacks. *** Piercing, Wood, Balloons, Chess piece, Paperclip, Metal token, Key, Erasers, Pen caps.

**Table 2 children-08-01182-t002:** Timing and features of endoscopy procedures.

Foreign Body	Symptoms	Localization	Hours from Ingestion	Removal
Button battery	No	Stomach	2	Yes
Button battery	No	Stomach	2	Yes
Food Bolus	Yes	Esophagus	3	Yes
Hair pin	No	Stomach	3	Yes
Shard of glass	Yes	Undefined	3	No
Shard of glass	No	Undefined	4	No
Clip hair (>2.5 cm)	Yes	Jejunum	8	No
Clip hair (3 cm)	No	Stomach	6	No
Coin	Yes	Esophagus	2	Yes
Coin	No	Esophagus	3	Yes
Coin	Yes	Esophagus	3	No
Coin	No	Esophagus	3	Yes
Coin	No	Mesogastrium	4	No
Metal object	Yes	Esophagus	3	Yes
Plastic object (4 cm)	Yes	Undefined	2	No
Piercing	No	Stomach	6	No
Screws	No	Duodenum	2	Yes

## Data Availability

No new data were created or analyzed in this study. Data sharing is not applicable to this article.

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
