# Peer review of "Foreign Body Ingestion in Children: Epidemiological, Clinical Features and Outcome in a Third Level Emergency Department"

_children, 2021, doi:10.3390/children8121182_

Round 1

Reviewer 1 Report

Thank you for the opportunity to read the manuscript “Foreign body ingestion in children: epidemiological, clinical features and outcome in a third level emergency department”.

The authors report a retrospective analysis of their single centre data on children with foreign body ingestions.

They start their manuscript with a profound and easy to read introduction which reports the most important statement and recommendations about this issue. The only statement I can not fully follow is in Line 40: “Mortality associated with the ingestion of foreign bodies is very rare.” There is a cause of death well known associated to battery ingestions by developing fistulae with access to the trachea or to large vessels with children bleeding to death. This is reported in several publications. The national poison control centre of the US reports actually 68 children with fatal events (https://www.poison.org/battery/fatalcases). I am very sure, not every case that occurs is detected and reported there and this number (solely for the US) underestimates the incidence. However, even the rest of the manuscript does not provide an adequate information about this very important issue.

The results miss to report dysphagia as the most common symptom, if a FB is lodged into the oesophagus. Did those cases reported with a proven localization of the FB in the oesophagus not suffer from dysphagia? This seems quite unlikely to me. We did a trial were 96% of such patients had dysphagia.

The rest of the manuscript is well written and all of the important issues are thoroughly addressed. I cannot see any novelty in the report, nevertheless addressing this issue in publications is worthful, especially if the manuscript provides a short review about it, like this one well does.

Author Response

Reviewer #1: The authors report a retrospective analysis of their single centre data on children with foreign body ingestions.

  1. They start their manuscript with a profound and easy to read introduction which reports the most important statement and recommendations about this issue. The only statement I can not fully follow is in Line 40: “Mortality associated with the ingestion of foreign bodies is very rare.” There is a cause of death well known associated to battery ingestions by developing fistulae with access to the trachea or to large vessels with children bleeding to death. This is reported in several publications. The national poison control centre of the US reports actually 68 children with fatal events (https://www.poison.org/battery/fatalcases). I am very sure, not every case that occurs is detected and reported there and this number (solely for the US) underestimates the incidence. However, even the rest of the manuscript does not provide an adequate information about this very important issue.

We modified this statement as suggested, introducing a citation about a review that expones this important problem in the lines 40-42:” Increasing prevalence of smaller, more technologically advanced toys in the household has resulted in an increased exposure to higher voltage batteries and powerful magnets that carry a high incidence of morbidity and mortality. [6]”

In our study there were not any case of such a severe complications, so we did not mention it in the results, but, according to your suggestion, we delete the statement in the conclusion in Line 240:” ..and severe and life-threatening complications are rare.”.

  1. The results miss to report dysphagia as the most common symptom, if a FB is lodged into the oesophagus. Did those cases reported with a proven localization of the FB in the oesophagus not suffer from dysphagia? This seems quite unlikely to me. We did a trial were 96% of such patients had dysphagia.

In our study 19 patients (6.6% of the total with symptoms) reported dysphagia; we called it “globus sensation or pharyngodynia”, because it is a specific sensation reported by patients and they often did not clarified it in the exact way. Even in our study, dysphagia was therefore one of the most frequently reported symptoms. We changed “Globus sensation” in “Disphagia” in the Table 1 

Reviewer 2 Report

The authors present a retrospective study examining foreign body ingestion in children in a third level emergency department. Overall the study is well conceived as it covers symptoms, foreign bodies ingested, need for imaging and endoscopic removal and guidelines. Overall edits for English are needed. Some edits are required to better clarify the scientific value of the manuscript:

Introduction:

Line  32: Please provide a citation for the age group at risk.

Line 40: Please provide a citation  for the statement on mortality associated with ingestion of foreign bodies.

Methods:

Line 61: More details are needed on how the data on the diagnosis of foreign body ingestion was abstracted. Was it done by code? If so , what codes were used to abstract the data?

Line 65: How were reports of instrumental examinations  and endoscopic procedures abstracted- by code? by reviewing the text of the procedure? Please specify. 

Results:

Line 83: Explain in what the codes "green" and "yellow" indicate

Table 1: Please add a footnote for the table to indicate what Foreign bodies includes. This is described in the results but needed near the table.

Discussion:

Line 187: Define "domestic observation"

Author Response

Dear Editor,

thank you very much for your letter of reply.

We have read the Reviewers’ comments and we revised our manuscript in accordance with their suggestions, hoping we did them in the most proper way.

This is a point-by-point list of replies to each Reviewers’ queries and changes made to our paper.

 Reviewer #2:

The authors present a retrospective study examining foreign body ingestion in children in a third level emergency department. Overall the study is well conceived as it covers symptoms, foreign bodies ingested, need for imaging and endoscopic removal and guidelines.

  1. Overall edits for English are needed.

We revised English language and highlighted corrections in red.

  1. Some edits are required to better clarify the scientific value of the manuscript:

Introduction:

-Line 32: Please provide a citation for the age group at risk.

We added a citation about the age group at risk: “Ingestion of foreign bodies is a common cause of access to the Emergency Room in the pediatric age, with up to 75% of cases occurring in children under 4 years of age. [1-2]”

-Line 40: Please provide a citation for the statement on mortality associated with ingestion of foreign bodies.

We delete this statement as suggested, introducing a citation about a review that expones this important problem in the lines 40-42:” Increasing prevalence of smaller, more technologically advanced toys in the household has resulted in an increased exposure to higher voltage batteries and powerful magnets that carry a high incidence of morbidity and mortality. [6]”

Methods:

-Line 61: More details are needed on how the data on the diagnosis of foreign body ingestion was abstracted. Was it done by code? If so, what codes were used to abstract the data?

We added this statement in the lines 68-72, as suggested: “Patients were identified from the hospital computerized clinical record (GIPSE®) by searching for the keywords "Foreign body ingestion", "Foreign body", "Foreign body obstruction" for all patients admitted to the ED.

Clinical and demographic data were collected by pediatric specialists after being trained in a data collection by a form developed specifically for the study”

-Line 65: How were reports of instrumental examinations and endoscopic procedures abstracted- by code? by reviewing the text of the procedure? Please specify. 

We added this statement in the lines 72-72, as suggested: “Reports of instrumental examinations and endoscopic procedures performed were extracted by reviewing the text of the procedure”

Results:

-Line 83: Explain in what the codes "green" and "yellow" indicate.

The green and yellow codes are the color codes proposed by the triage nurses upon the patient's arrival in the Emergency Room and refer to the urgency of the examination of the child. We added in the text these sentences: “Referring to the urgency of the patients’ examination  at the triage 65% (187) were admitted with a green code (uncritical patient, low priority access to care), 35% (99) with a yellow code (moderately critical patient, quick access to treatment).  

-Table 1: Please add a footnote for the table to indicate what Foreign bodies includes. This is described in the results but needed near the table.

We added a Footnote to the Table 1 to indicate what Foreign bodies includes:

*Coins, Marbles, Beads, Jewelry (earrings, rings, pendants), Hair clips, Rubber objects, Paper and cardboard

**Fish bones, Plastic objects / fragments, Shards of glass, Metal objects / fragments, Toothpick, Aug, Screws and nails, Thumbtacks

*** Piercing, Wood, Ballons, Chess piece, Paperclip, Metal token, Key, Erasers, Pen caps.”

Discussion:

-Line 187: Define "domestic observation"

In asymptomatic patients with gastric or intestinal foreign bodies and no elements of risk, home observation is generally indicated.

Home observation must be carried out by a parent or caregiver of the child and consists in carefully checking the child's feces at each evacuation, in order to find the foreign body.

Thanking you again for this precious opportunity of cooperation and hoping to receive as soon as possible your final decision, we send to you our personal best regards,

Antonio Gatto & co-write
